EMBO
Molecular Medicine

# Pathogen-specific B-cell receptors drive chronic lymphocytic leukemia by light-chain-dependent cross-reaction with autoantigens

Nereida Jiménez de Oya[1,†], Marco De Giovanni[1,2,†], Jessica Fioravanti[1,‡], Rudolf Übelhart[3,‡], Pietro Di Lucia[1], Amleto Fiocchi[1], Stefano Iacovelli[4], Dimitar G Efremov[4], Federico Caligaris-Cappio[5,§], Hassan Jumaa[3,6] (iD), Paolo Ghia[2,5], Luca G Guidotti[1] & Matteo Iannacone[1,2,7,*] (iD)

## Abstract

Several lines of evidence indirectly suggest that antigenic stimulation through the B-cell receptor (BCR) supports chronic lymphocytic leukemia (CLL) development. In addition to self-antigens, a number of microbial antigens have been proposed to contribute to the selection of the immunoglobulins expressed in CLL. How pathogen-specific BCRs drive CLL development remains, however, largely unexplored. Here, we utilized mouse models of CLL pathogenesis to equip B cells with virus-specific BCRs and study the effect of antigen recognition on leukemia growth. Our results show that BCR engagement is absolutely required for CLL development. Unexpectedly, however, neither acute nor chronic exposure to virus-derived antigens influenced leukemia progression. Rather, CLL clones preferentially selected light chains that, when paired with virus-specific heavy chains, conferred B cells the ability to recognize a broad range of autoantigens. Taken together, our results suggest that pathogens may drive CLL pathogenesis by selecting and expanding pathogen-specific B cells that cross-react with one or more self-antigens.

**Keywords** autoantigens; B-cell receptor; chronic lymphocytic leukemia; infection; light chains

**Subject Categories** Cancer; Haematology; Immunology

## Introduction

Chronic lymphocytic leukemia (CLL), the most common adult leukemia in the Western World, is characterized by the clonal expansion of CD5[+] B cells in blood and peripheral tissues (Zhang & Kipps, 2014). BCR signaling plays a critical role in CLL pathogenesis (Burger & Chiorazzi, 2013), and, accordingly, inhibitors targeting BCR-associated kinases [Bruton's tyrosine kinase (BTK), phosphoinositide 3-kinase (PI3K) δ] have shown great clinical efficacy in patients (Burger & Chiorazzi, 2013; Furman *et al*, 2014).

Patient-derived CLL cells with either unmutated or mutated immunoglobulin genes often express similar, if not identical, BCRs with common stereotypic features and/or structural similarities (Burger & Chiorazzi, 2013; Stevenson *et al*, 2014; Zhang & Kipps, 2014). This marked restriction in the immunoglobulin gene repertoire of CLL cells suggests that binding to restricted sets of antigenic epitopes is key to the selection and the expansion of those normal B-cell clones that eventually enter the CLL pathogenic process (Burger & Chiorazzi, 2013; Stevenson *et al*, 2014; Zhang & Kipps, 2014). The nature of such antigens and the mechanisms of BCR stimulation during CLL remain, however, incompletely understood. The majority of unmutated CLLs express low-affinity BCRs that are polyreactive to several autoantigens (Burger & Chiorazzi, 2013; Stevenson *et al*, 2014; Zhang & Kipps, 2014). The BCR specificity of mutated CLLs is less characterized, although a number of self-antigens, as well as microbial or virus-associated antigens, have been identified (Zhang & Kipps, 2014). Indeed, epidemiological studies indicate that several infections are associated with CLL development and CLL-associated immunoglobulins are known to react with various viruses and other pathogens (Landgren *et al*, 2007; Lanemo

1 Division of Immunology, Transplantation and Infectious Diseases, IRCCS San Raffaele Scientific Institute, Milan, Italy
2 Vita-Salute San Raffaele University, Milan, Italy
3 Institute of Immunology, University Hospital Ulm, Ulm, Germany
4 Molecular Hematology Unit, International Centre for Genetic Engineering & Biotechnology, Trieste, Italy
5 Division of Experimental Oncology, IRCCS San Raffaele Scientific Institute, Milan, Italy
6 Department of Molecular Immunology, Faculty of Biology, Albert-Ludwigs University of Freiburg, Freiburg, Germany
7 Experimental Imaging Center, IRCCS San Raffaele Scientific Institute, Milan, Italy
*Corresponding author. Tel: +39 (02) 2643 6359; E-mail: iannacone.matteo@hsr.it
†These authors contributed equally to this work as first authors
‡These authors contributed equally to this work as second authors
§Present address: Associazione Italiana Ricerca sul Cancro, Milan, Italy

Myhrinder *et al*, 2008; Kostareli *et al*, 2009; Steininger *et al*, 2009, 2012; Hoogeboom *et al*, 2013; Hwang *et al*, 2014). If and how pathogen-specific BCRs drive CLL development and progression is largely unexplored.

## Results and Discussion

To begin addressing these issues, we took advantage of a well-established CLL mouse model, the Eµ-TCL1 transgenic mouse, where the oncogene *Tcl1* is expressed in both immature and mature B cells (Bichi *et al*, 2002). As such, Eµ-TCL1 transgenic mice develop a lymphoproliferative disorder that entails the clonal expansion of $CD5^+$ $IgM^+$ B cells (Bichi *et al*, 2002). Like in human CLL, the immunoglobulin rearrangements from different Eµ-TCL1 leukemic mice can be structurally very similar and closely resemble antibodies reactive to self and to microbial antigens (Yan *et al*, 2006). We started out by generating Eµ-TCL1 mice that either expressed defined virus-specific BCRs or that lacked the BCR entirely. To this end, we bred Eµ-TCL1 mice against the following mouse lineages: (i) KL25 mice (Hangartner *et al*, 2003), which carry a gene-targeted immunoglobulin heavy chain expressing a neutralizing specificity for lymphocytic choriomeningitis virus (LCMV) strain WE; (ii) VI10YEN mice (Hangartner *et al*, 2003), which carry a gene-targeted immunoglobulin heavy chain (VI10) and a transgenic non-targeted immunoglobulin light chain (YEN) expressing a neutralizing specificity for vesicular stomatitis virus (VSV) serotype Indiana; and (iii) $D_H$LMP2A mice (Casola *et al*, 2004), which carry a targeted replacement of *Igh* by the Epstein–Barr virus protein LMP2A and develop B cells lacking surface-expressed and secreted immunoglobulins. Of note, the choice of these particular virus-specific transgenic BCR lineages rests on the notion that they have been extremely useful at characterizing the role of humoral immunity in the pathogenesis of acute and chronic viral infections (Hangartner *et al*, 2003, 2006; Sammicheli *et al*, 2016) and that, being heavy-chain knock-in lineages, they allow for examining the eventual role of light chains in CLL development.

B cells isolated from the resulting progeny were first characterized with regard to *Tcl1* expression, subset development, and BCR responsiveness. As shown in Fig EV1A, 8-week-old pre-leukemic KL25 × Eµ-TCL1, VI10YEN × Eµ-TCL1, and $D_H$LMP2A × Eµ-TCL1 mice showed levels of splenic *Tcl1* expression that were comparable to those of Eµ-TCL1 mice expressing a polyclonal BCR repertoire. We next enumerated $CD5^+$ B cells in blood, serosal cavities, secondary lymphoid organs and liver of the above-mentioned mouse strains, again at 8 weeks of age. The number of $CD5^+$ B cells in blood, spleen and lymph nodes of KL25 × Eµ-TCL1,

VI10YEN × Eµ-TCL1, and $D_H$LMP2A × Eµ-TCL1 mice was similar to that detected in the same districts of Eµ-TCL1 mice (Fig EV1B and C); performing a similar comparison in liver and peritoneum, however, indicated that $CD5^+$ B cells were reduced in KL25 × Eµ-TCL1, VI10YEN × Eµ-TCL1, and $D_H$LMP2A × Eµ-TCL1 mice (Fig EV1B and C). While the molecular basis for this selective reduction of $CD5^+$ B cells in liver and peritoneum of pre-leukemic KL25 × Eµ-TCL1, VI10YEN × Eµ-TCL1, and $D_H$LMP2A × Eµ-TCL1 mice are unclear, it is worth noting that the extent of such reduction was similar in all the 3 above-mentioned mouse lineages (Fig EV1B and C). We finally confirmed that both KL25 × Eµ-TCL1 and VI10YEN × Eµ-TCL1 mice express the respective virus-specific BCR in all B-cell subsets and appropriately respond to cognate antigen stimulation (Fig EV1D–G).

We next compared KL25 × Eµ-TCL1, VI10YEN × Eµ-TCL1, and $D_H$LMP2A × Eµ-TCL1 mice to Eµ-TCL1 mice expressing a polyclonal BCR repertoire with regard to leukemia development at steady state (in the absence of cognate antigen challenge). Disease progression was monitored by quantifying the frequency of $CD5^+$ B cells in peripheral blood (Fig 1A), and we arbitrarily defined leukemic those mice that had ≥ 20% $CD5^+$ cells among total $CD19^+$ B cells (a frequency never reached by WT mice in the 36-week-long observation period) (Fig 1B). As previously reported (Bichi *et al*, 2002), Eµ-TCL1 mice showed an expansion of circulating $CD5^+$ B cells as early as 16 weeks of age, and by 36 weeks of age 100% of them were frankly leukemic, with accumulation of large numbers of $CD5^+$ B cells in blood, serosal cavities and lymphoid as well as non-lymphoid organs (Fig 1A–C). $D_H$LMP2A × Eµ-TCL1 mice showed a profound impairment in leukemia development (Fig 1A–C), indicating that BCR expression is required for leukemia growth and that the tonic signal provided by the LMP2A protein is not sufficient to support leukemic expansion. We then assessed whether pathogen-specific BCRs sustained cancer development. Both KL25 × Eµ-TCL1 and VI10YEN × Eµ-TCL1 mice developed CLL, even though they differed in regard to disease incidence and leukemic cell accumulation. Whereas CLL development in KL25 × Eµ-TCL1 mice occurred at a rate that was indistinguishable from that of Eµ-TCL1 mice, VI10YEN × Eµ-TCL1 mice had a more indolent course of disease (Fig 1A–C). These results indicate that the BCR shapes CLL incidence and behavior *in vivo*.

Before attempting to pinpoint the mechanisms underlying the difference in CLL incidence between KL25 × Eµ-TCL1 and VI10YEN × Eµ-TCL1 mice, we sought to determine whether cognate antigen recognition, prior to disease onset, influenced CLL development and progression in Eµ-TCL1 mice expressing virus-specific BCRs. To this end, we infected 8-week-old VI10YEN × Eµ-TCL1 mice (and Eµ-TCL1 controls) with $10^6$ p.f.u. of VSV Indiana. VSV

---

**Figure 1.  Leukemia development in KL25 × Eµ-TCL1, VI10YEN × Eµ-TCL1, and $D_H$LMP2A × Eµ-TCL1 mice.**

A   Percentage of $CD5^+$ cells (out of total $CD19^+$ peripheral blood leukocytes) in WT (gray), Eµ-TCL1 (black), KL25 × Eµ-TCL1 (red), VI10YEN × Eµ-TCL1 (blue), and $D_H$LMP2A × Eµ-TCL1 (green) male mice at the indicated time points. Two-way ANOVA (Bonferroni's multiple comparison).

B   Incidence of leukemia (defined as ≥ 20% $CD5^+$ cells out of total $CD19^+$ peripheral blood leukocytes) over time in the same mice described in (A). n = 4–20 (WT), 16–45 (Eµ-TCL1), 8–29 (KL25 × Eµ-TCL1), 14–34 (VI10YEN × Eµ-TCL1), 19–35 ($D_H$LMP2A × Eµ-TCL1). Log-rank (Mantel-Cox).

C   Percentage of $CD5^+$ cells (out of total $CD19^+$ cells) in the indicated organs of the same mice described in (A) at 36 weeks of age. n = 6 (WT), 9 (Eµ-TCL1), 6 (KL25 × Eµ-TCL1), 8 (VI10YEN × Eµ-TCL1), 12 ($D_H$LMP2A × Eµ-TCL1). One-way ANOVA (Bonferroni's multiple comparison).

Data information: Results are expressed as mean + SEM. *P < 0.05, **P < 0.01, ***P < 0.001. Exact P-values for each experiment are reported in Appendix Table S1.
Source data are available online for this figure.

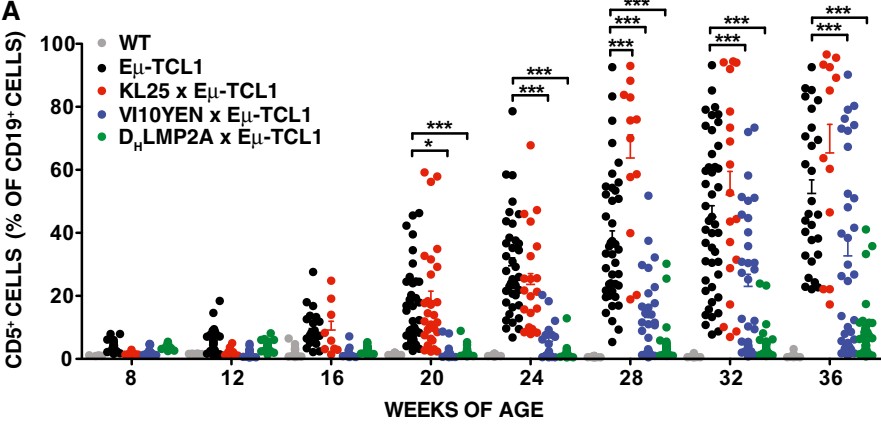

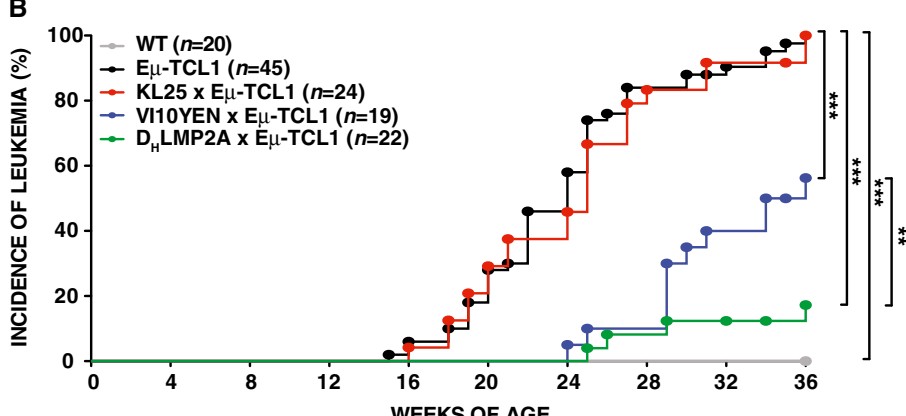

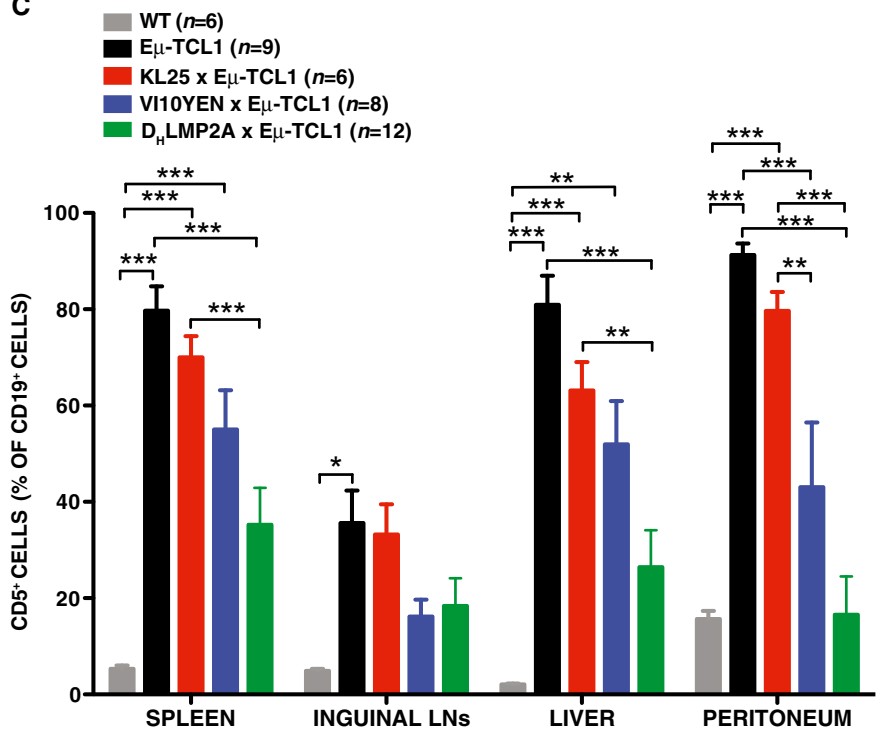

Figure 1.

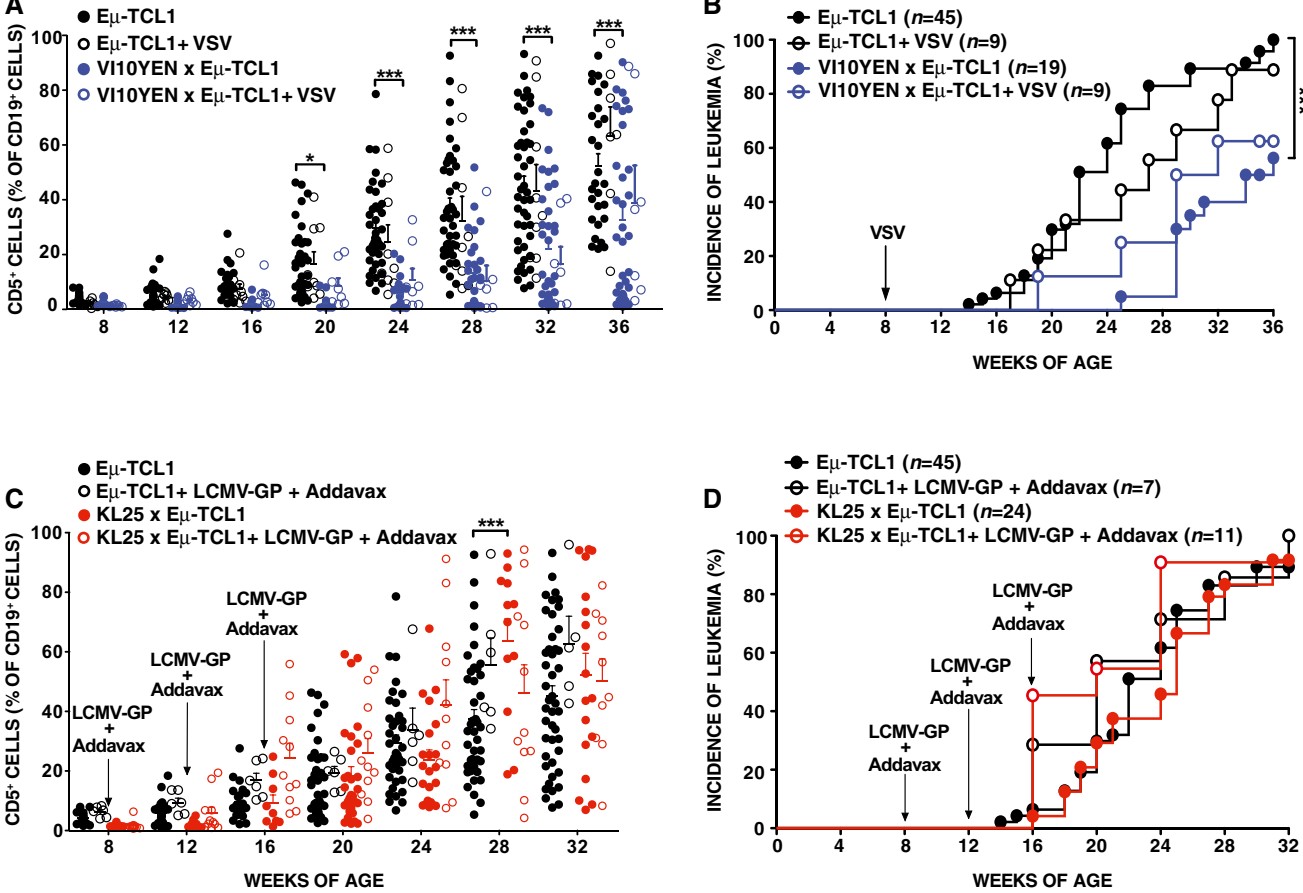

**Figure 2. High-affinity antigen recognition does not affect CLL development or progression.**

A   Percentage of CD5⁺ cells (out of total CD19⁺ peripheral blood leukocytes) over time in Eµ-TCL1 (black) and VI10YEN × Eµ-TCL1 (blue) male mice that were infected (open symbols) or not (closed symbols) with $10^6$ p.f.u. of VSV Indiana at 8 weeks of age.

B   Incidence of leukemia (defined as ≥ 20% CD5⁺ cells out of total CD19⁺ peripheral blood leukocytes) over time in the same mice described in (A). n = 16–45 (Eµ-TCL1), 9 (Eµ-TCL1 + VSV), 14–34 (VI10YEN × Eµ-TCL1), 7–9 (VI10YEN × Eµ-TCL1 + VSV).

C   Percentage of CD5⁺ cells (out of total CD19⁺ peripheral blood leukocytes) over time in Eµ-TCL1 (black) and KL25 × Eµ-TCL1 (red) mice that were immunized (open symbols) or not (closed symbols) with LCMV-GP + Addavax at the indicated time points.

D   Incidence of leukemia (defined as ≥ 20% CD5⁺ cells out of total CD19⁺ peripheral blood leukocytes) over time in the same mice described in (C). n = 16–45 (Eµ-TCL1), 7 (Eµ-TCL1 + LCMV-GP + Addavax), 8–29 (KL25 × Eµ-TCL1), 10–11 (KL25 × Eµ-TCL1 + LCMV-GP + Addavax).

Data information: Results are expressed as mean + SEM. *$P < 0.05$, ***$P < 0.001$. Exact $P$-values for each experiment are reported in Appendix Table S1. Two-way ANOVA (Bonferroni's multiple comparison) was used in (A, C); Log-Rank (Mantel–Cox) was used in (B, D).
Source data are available online for this figure.

infection induced B cells in VI10YEN × Eµ-TCL1 mice to get activated, proliferate, and differentiate into Ab-secreting cells (Fig EV2A), but it did not alter the kinetics of leukemia development or progression (Figs 2A and B, and EV2B). We then infected 8-week-old KL25 × Eµ-TCL1 mice and Eµ-TCL1 controls with $10^6$ f.f.u. of LCMV WE. Unexpectedly, LCMV infection of both KL25 × Eµ-TCL1 mice and Eµ-TCL1 controls (that express a polyclonal BCR repertoire) abrogated CLL development (Fig EV3). The cellular and molecular underpinnings of this intriguing observation extend beyond the scope of this study and will be the subject of a future report. To avoid potentially confounding effects, we henceforth decided to test the role of antigenic stimulation in this setting by repetitively immunizing 8-week-old KL25 × Eµ-TCL1 mice (and Eµ-TCL1 controls) with the purified LCMV WE glycoprotein [which contains the antigenic determinant recognized by KL25 B cells

(Sammicheli *et al*, 2016)]. Although LCMV immunization induced B cells in KL25 × Eµ-TCL1 mice to get activated, proliferate and differentiate into Ab-secreting cells (Fig EV4A), it did not alter the kinetics of leukemia development or progression (Figs 2C and D, and EV4B and C). Together, these results suggest that high-affinity recognition of pathogen-derived antigens does not affect CLL development or progression, and they prompted us to investigate whether virus-specific BCRs may drive CLL pathogenesis by mechanisms that are unrelated to pathogen specificity.

To begin investigating such potential pathogenic mechanisms, we analyzed the BCR repertoire of leukemic KL25 × Eµ-TCL1 and VI10YEN × Eµ-TCL1 mice and compared it to that of age-matched KL25 and VI10YEN mice. Since both KL25 and VI10YEN are knock-in for the BCR heavy chain (Hangartner *et al*, 2003), there is no alternative heavy chain that these mice can express. Accordingly, all

analyzed leukemic KL25 × Eμ-TCL1 and VI10YEN × Eμ-TCL1 mice expressed the expected transgenic heavy chain as an IgM (Tables EV1 and EV2, and Fig EV5). As per the light-chain repertoire, we noticed that, when compared with age-matched KL25 and VI10YEN mice, leukemic KL25 × Eμ-TCL1 and VI10YEN × Eμ-TCL1 mice had a biased light-chain usage. Specifically, among the light chains preferentially expressed by leukemic KL25 × Eμ-TCL1 mice, we found the IGKV12-44*01 F/IGKJ2*01 F gene associated with the LCDR3 motif CQH-HYGTPY-TF and the IGKV6-32*01 F/IGKJ2*01 F gene associated with the LCDR3 motif CQQ-DYSS-TF (Fig 3A and Table EV1). Similarly, leukemic VI10YEN × Eμ-TCL1 mice preferentially expressed the IGKV6-32*01 F/IGKJ2*01 F gene associated with the LCDR3 motif CQQ-DYSS-TF and the IGKV6-32*01 F/IGKJ2*01 F gene associated with the LCDR3 motif CQQ-DYSSPY-TF (the YEN transgenic light chain, Fig 3A and Table EV2). This preferential light-chain usage is reminiscent to what has been described for polyclonal Eμ-TCL1 (Yan *et al*, 2006) and suggested that leukemic KL25 × Eμ-TCL1 and VI10YEN × Eμ-TCL1 mice selected BCRs capable of cross-reacting with one or more autoantigens. One particular case in point is the capacity of CLL-derived BCRs to recognize an internal epitope of the BCR itself, a feature referred to as cell autonomous signaling (Dühren-von Minden *et al*, 2012). We therefore set out to test whether BCRs derived from leukemic KL25 × Eμ-TCL1 and VI10YEN × Eμ-TCL1 mice possessed cell autonomous signaling activity. To this end, we introduced the corresponding heavy and light chains in the BCR-deficient murine B-cell line TKO, which expresses an inactive B-cell linker (BLNK) adaptor protein that becomes functional in the presence of 4-hydroxytamoxifen (4-OHT) (Dühren-von Minden *et al*, 2012). Addition of 4-OHT to TKO cells with an autonomously active BCR results in signal activation and propagation, ultimately resulting in an increase in intracellular $Ca^{++}$ levels that is detectable by flow cytometry (Dühren-von Minden *et al*, 2012; Fig 3B). The BCR composed of the KL25 transgenic heavy chain coupled with the light chain that was most frequently expressed in non-leukemic 9-month-old KL25 mice (IGKV3-10*01 F/IGKJ1*01 F gene associated with the LCDR3 motif CQQ-NNEDPW-TF) was not autonomously active (Fig 3C, left panel); similarly, the BCR composed of the VI10 transgenic heavy chain coupled to the transgenic YEN light chain (the most frequently expressed light chain in non-leukemic 9-month-old VI10YEN mice) did not possess autonomous signaling capability (Fig 3D, left panel). We next evaluated BCRs composed of the same KL25 and VI10 heavy chains coupled to the light chains that were most frequently expressed in 9-month-old leukemic KL25 × Eμ-TCL1 and VI10YEN × Eμ-TCL1 mice, respectively. When expressed together with the KL25 heavy chain, two of the light chains that were most frequently selected in leukemic KL25 × Eμ-TCL1 mice endowed TKO

cells with autonomous signaling activity (Fig 3C, right panels). By contrast, when expressed together with the VI10 heavy chain, the light chain most frequently selected in leukemic VI10YEN × Eμ-TCL1 mice did not confer cell autonomous signaling activity to TKO cells (Fig 3D, right panel). The observation that KL25 × Eμ-TCL1 mice—which progress rapidly to CLL (Fig 1A and B)—show cell autonomous signaling activity, whereas VI10YEN × Eμ-TCL1 mice—which have a more indolent course of disease (Fig 1A and B)—do not is in agreement with the notion that autonomous signaling is an important pathogenic driver in CLL (Dühren-von Minden *et al*, 2012; Iacovelli *et al*, 2015), but argues against autonomous signaling being an absolute prerequisite of leukemia development. Future studies should assess whether patient-derived CLL cells that express a pathogen-reactive BCR possess or lack cell autonomous signaling capability and whether this property relates to disease activity. The observation that leukemic VI10YEN × Eμ-TCL1 mice showed a biased light-chain usage that did not confer autonomous signaling activity (Fig 3D) prompted us to explore the possibility that these leukemic BCRs might be cross-reacting with different autoantigens. To test this hypothesis and to characterize the nature of such potential autoantigens, we screened sera from young and old KL25, VI10YEN, Eμ-TCL1, KL25 × Eμ-TCL1, and VI10YEN × Eμ-TCL1 mice against a panel of 124 nuclear, cytoplasmic, membrane, and phospholipid autoantigens known to be targeted by autoantibodies in various autoimmune diseases and in mouse and human CLL (Yan *et al*, 2006; Catera *et al*, 2008) (see the complete list of antigens in Table EV3). Sera from 8-week-old KL25, VI10YEN, Eμ-TCL1, KL25 × Eμ-TCL1, and VI10YEN × Eμ-TCL1 mice as well as 9-month-old KL25 and VI10YEN mice did not react with most of the tested autoantigens; by contrast, sera from 9-month-old leukemic Eμ-TCL1, KL25 × Eμ-TCL1 and VI10YEN × Eμ-TCL1 mice bound avidly to the vast majority of autoantigens that were examined (Fig 3E). To formally demonstrate that autoantibodies were indeed produced by the malignant cells, we generated monoclonal IgMs bearing the KL25 heavy chain coupled to the IGKV12-44*01 F/IGKJ2*01 F light chain associated with the LCDR3 motif CQH-HYGTPY-TF (the most frequently selected light chain in leukemic KL25 × Eμ-TCL1 mice) or the VI10 heavy chain coupled to the IGKV6-32*01 F/IGKJ2*01 F light chain associated with the LCDR3 motif CQQ-DYSS-TF (the most frequently selected light chain in leukemic VI10YEN × Eμ-TCL1 mice). Both monoclonal antibodies showed a significant degree of autoreactivity, with the antibody derived from KL25 × Eμ-TCL1 mice recognizing a broader range of autoantigens (Fig 3E). These results indicate that leukemic KL25 × Eμ-TCL1 and VI10YEN × Eμ-TCL1 mice (and, possibly, polyclonal Eμ-TCL1 mice) preferentially selected light chains that confer BCRs the capacity to cross-react with a broad range of autoantigens. The data are consistent with the

**Figure 3.  Pathogen-specific B-cell receptors drive chronic lymphocytic leukemia by light-chain-dependent cross-reaction with autoantigens.**

A    Pie charts representing LCDR3 usage in 9-month-old KL25 (top left), VI10YEN (top right), KL25 × Eμ-TCL1 (bottom left), and VI10YEN × Eμ-TCL1 male mice (bottom right). *n* = 3 (KL25 and VI10YEN), 11 (KL25 × Eμ-TCL1), 12 (VI10YEN × Eμ-TCL1).

B–D    Representative flow cytometry analyses of $Ca^{2+}$ flux after activation of the ERT2-BLNK fusion protein by 4-OHT with or without an anti-mouse light-chain antibody (α-BCR) in TKO cells expressing the indicated BCR (the CDR3 corresponding to the expressed light chain is indicated in parentheses). Addition of 4-OHT with or without α-BCR is marked by an arrow. Results are representative of three independent experiments.

E    Sera from 8-week- and 9-month-old KL25, VI10YEN, Eμ-TCL1, KL25 × Eμ-TCL1, and VI10YEN × Eμ-TCL1 male mice or monoclonal IgMs from leukemic KL25 × Eμ-TCL1 and VI10YEN × Eμ-TCL1 mice were screened for the presence of autoantibodies against a panel of 124 nuclear, cytoplasmic, membrane, and phospholipid autoantigens. The heat maps are based on the normalized fluorescent intensity of autoantibodies and are represented on a color scale range between +200 (red) and −200 (green) standard deviations.

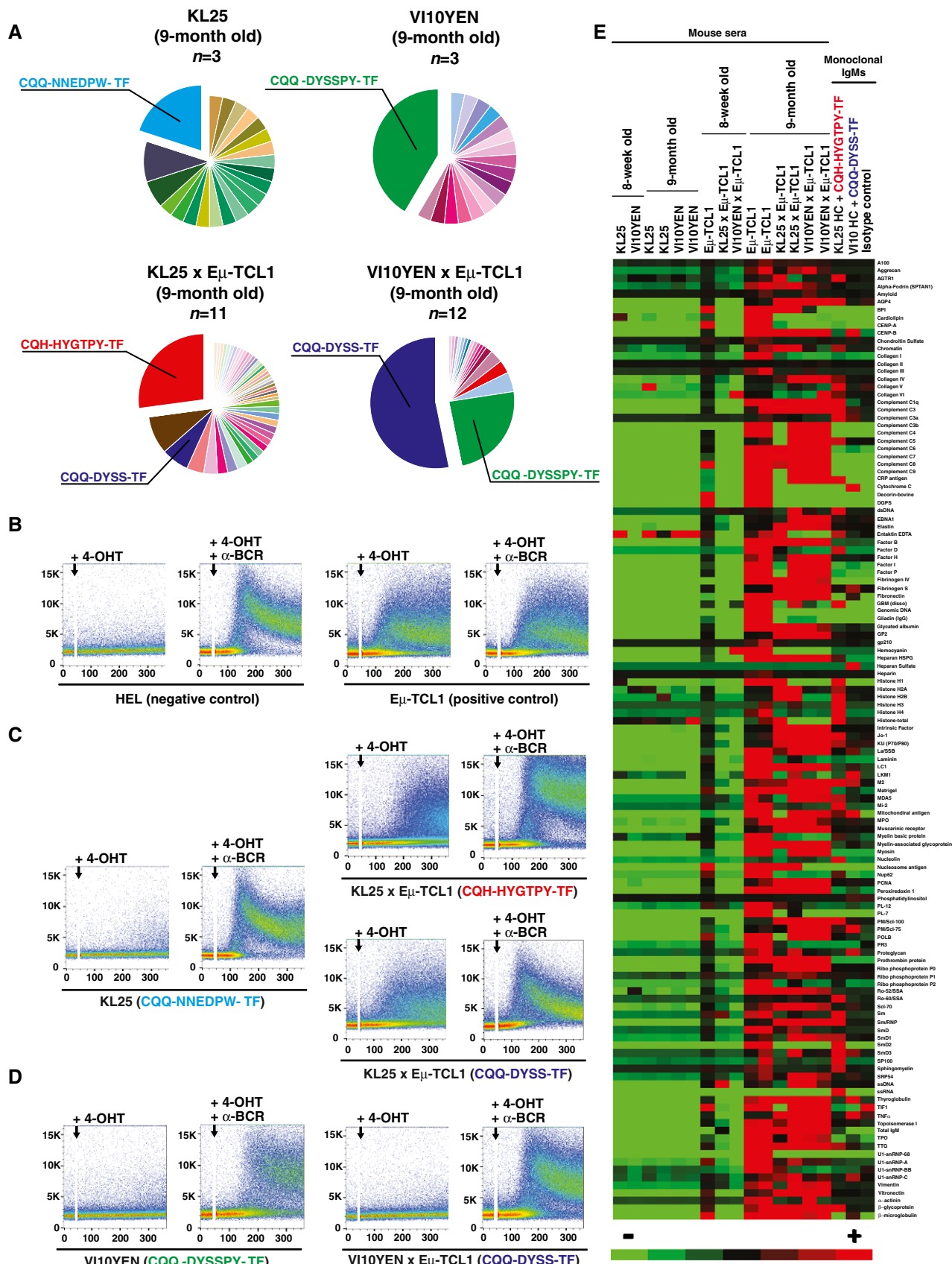

**Figure 3.**

hypothesis, based on evidence obtained in CLL patients, that light chains, in association with defined heavy chains, are crucial in shaping the specificity of leukemic BCRs (Stamatopoulos et al, 2005; Hadzidimitriou et al, 2009; Kostareli et al, 2010).

In conclusion, we here demonstrate that BCR expression is required for leukemia growth in the Eμ-TCL1 transgenic mouse model and that CLL clones preferentially select light chains that, when paired with virus-specific heavy chains, confer B cells the ability to recognize a broad range of autoantigens. Taken together, our results suggest that pathogens may drive CLL pathogenesis by selecting and expanding pathogen-specific B cells that cross-react with one or more self-antigens. The interaction between malignant B cells and self-antigens may then be crucial for disease progression.

# Materials and Methods

### Mice

C57BL/6 and CD45.1 (inbred C57BL/6) mice were purchased from Charles River. Eμ-TCL1 mice (Bichi et al, 2002) were provided by C. Croce (Ohio State University). $D_H$LMP2A mice (Casola et al, 2004) (inbred Balb/c) were originally provided by K. Rajewsky (Harvard Medical School) and bred more than 10 generations against C57BL/6 mice. Heavy-chain knock-in and light-chain BCR-transgenic mice specific for VSV Indiana (VI10YEN (Hangartner et al, 2003)) and heavy-chain knock-in BCR-transgenic mice specific for LCMV WE (KL25, Hangartner et al, 2003) were obtained through the European Virus Archive. Of note, the Eμ-TCL1 transgene was brought to homozygosity in all lineages. Mice were housed under specific pathogen-free conditions, and, in all experiments, they were matched for age and sex before experimental manipulation. All experimental animal procedures were approved by the Institutional Animal Committee of the San Raffaele Scientific Institute.

### Infections and immunizations

Eight-week-old male mice were infected intravenously with $1 \times 10^6$ plaque-forming units (p.f.u.) of VSV serotype Indiana, or with $1 \times 10^6$ focus-forming units (f.f.u.) of LCMV WE. Alternatively, mice were immunized intramuscularly with 10 μg of recombinant LCMV WE glycoprotein (GP)-1-human IgG fusion protein (Sommerstein et al, 2015) mixed at 1:1 ratio with a squalene-based oil-in-water nano-emulsion (AddaVax, InvivoGen), as described (Sammicheli et al, 2016). Viruses were propagated and quantified as described (Iannacone et al, 2008; Tonti et al, 2013), and dissolved in 200 μl of PBS prior to intravenous injection.

Mice were retro-orbitally bled at the indicated time points for VSV- or LCMV-specific Abs measured by VSV neutralization assay or LCMV focus reduction assay, as described (Sammicheli et al, 2016). All infectious work was performed in designated BSL-2 and BSL-3 workspaces in accordance with institutional guidelines.

### Flow cytometry-based analyses

Mice were bled retro-orbitally every 2 weeks, and leukemia development and progression were assessed by flow cytometry-based quantification of $CD19^+$ $CD5^+$ cells. White blood cell counts were performed on an automated cell counter (HeCoVet; Seac–Radim). Mice were sacrificed at 36 weeks of age or earlier if they showed signs of advanced leukemia (i.e. lethargy, impaired mobility, hunched posture, labored breathing, splenomegaly, and/or hepatomegaly). Single-cell suspensions of livers, spleens, and lymph nodes were generated as described (Tonti et al, 2013; Guidotti et al, 2015). Peritoneal cells were removed by injection of 10 ml of HBSS into the peritoneal cavity followed by withdrawal of the peritoneal exudates. All flow cytometry stainings of surface-expressed molecules were performed as described (Sammicheli et al, 2016). Antibodies used included PB-conjugated anti-CD19 (eBio1D3, BD Pharmingen), APC- and PerCP-conjugated anti-CD5 (53–7.3, BD Pharmingen), FITC-conjugated anti-CD23 (B3B4, BD Pharmingen), PE-Cy7-conjugated anti-CD3 (145-2C11, BD Pharmingen), APC-conjugated anti-IgM (11/41, BD Pharmingen), Alexa-Fluor 546 anti-mouse IgG (polyclonal, Invitrogen), FITC-conjugated anti-CD69 (H1.2F3, BD Pharmingen), PE-conjugated anti-CD25 (PC61, BioLegend), eFluor 450-conjugated anti-B220 (RA3-682, eBioscience), PerCP-conjugated anti-B220 (RA3-B2, BioLegend), and PE-conjugated Streptavidin (BD Pharmingen). The anti-idiotypic antibodies 35.61 (Hangartner et al, 2003) (which recognizes a combinatorial determinant provided by $V_H$ and $V_k$ of VI10) and III-C4.8 (Hangartner et al, 2003) (which recognizes the $V_H$ of KL25) were produced from hybridoma supernatants and biotinylated according to standard methods. All flow cytometry analyses were performed in FACS buffer containing PBS with 2 mM EDTA and 2% FBS on a FACS CANTO (BD Pharmingen) and analyzed with FlowJo software (Treestar Inc.).

### Quantification of TCL1 expression

Total RNA was extracted from splenocytes using the ReliaPrep$^{tm}$ kit (Promega) according to the manufacturer's instructions. mRNA was reverse-transcribed with Promega RT reagents (Promega). Real-time quantitative PCRs were performed on a ABI 7900 HT Fast Real-Time PCR System (Applied Biosystems) with FastStart Universal SYBR Green Master Mix (Roche) using the following primers: TCL-1 (forward) 5′-GCCTGGCTGCCCTTAACC-3′, TCL-1 (reverse) 5′-GACGCA AGAGCACCCGTAAC-3′, β-actin (forward) 5′-AAGAGAAGGGTTACC CGGGATA-3′, β-actin (reverse) 5′-CCTAAGGCCAACCGTGAAAA-3′. Every reaction was run in triplicates, and β-actin levels were used as an endogenous control for normalization.

### B-cell activation *in vitro*

Naïve B cells from the spleens of KL25 × Eμ-TCL1 and VI10YEN × Eμ-TCL1 were negatively selected by magnetic isolation and tested for their capacity to get activated and proliferate in response to PFA-inactivated VSV or LCMV as described (Sammicheli et al, 2016).

### IGHV and IGLV sequencing analysis

Total cellular RNA was isolated from the spleens of leukemic mice using ReliaPrep™ RNA Tissue Miniprep System (Promega). RNA was reverse-transcribed using oligo-dT, and cDNA was amplified by PCR using Phusion® Flash High-Fidelity PCR Master Mix (Thermo Fisher Scientific) with FR1 and constant region primers (Iacovelli et al, 2015). PCR products were purified using Wizard® SV Gel and

**The paper explained**

**Problem**

Chronic lymphocytic leukemia (CLL), the most common adult leukemia in the Western World, is characterized by the clonal expansion of a subset of B lymphocytes. Epidemiological studies have associated several infections to CLL, but if and how pathogens drive leukemia development and progression is largely unexplored.

**Results**

We show here, in animal models of disease, that a specific protein (the B-cell receptor) is required for leukemia development. Moreover, acute or chronic infections do not lead to faster leukemia development or progression. Rather, recognition of autoantigens is a major pathogenic driver in CLL.

**Impact**

Our results help clarify the role of B-cell receptor signaling in CLL and should instruct the further development and use of drugs targeting this pathway.

PCR Clean-Up System (Promega) and directly sequenced using Sanger ABI 3730xl (GATC Biotech). Samples with more than one IGHV or IGLV rearrangement were cloned into Zero Blunt® TOPO® plasmid and analyzed by sequencing. Alignments of IGHV or IGLV genes against murine germline sequences were performed with IMGT/V-QUEST software (http://www.imgt.org/).

### Analysis of cell autonomous BCR signaling activity

IGHV and IGLV genes were amplified by anchor-PCR using poly-G-tailed complementary DNA and a poly-C-containing primer and inserted into respective retroviral vectors, as described (Dühren-von Minden *et al*, 2012). $1 \times 10^6$ freshly transduced TKO cells expressing ERT2–SLP65 was loaded with Indo-1 (Invitrogen) using Pluronic (Invitrogen). Induction of ERT2–SLP65 was performed by the addition of 2 μM 4-OHT (Sigma-Aldrich). Calcium flux was measured with LSR Fortessa (Becton Dickinson). Cross-linking of the BCR with goat anti-mouse kappa (10 μg/ml; Southern Biotech) was used as a positive control.

### Monoclonal IgM production

Recombinant monoclonal IgMs expressing the KL25 heavy chain coupled to the IGKV12-44*01 F/IGKJ2*01 F light chain associated with the LCDR3 motif CQH-HYGTPY-TF (the most frequently selected light chain in leukemic KL25 × Eμ-TCL1 mice) or the VI10 heavy chain coupled to the IGKV6-32*01 F/IGKJ2*01 F light chain associated with the LCDR3 motif CQQ-DYSS-TF (the most frequently selected light chain in leukemic VI10YEN × Eμ-TCL1 mice) were produced in HEK293 cells by Absolute Antibody (Oxford, UK).

### Protein microarray analysis

Sera from young and old KL25, VI10YEN, Eμ-TCL1, KL25 × Eμ-TCL1, and VI10YEN × Eμ-TCL1 mice and monoclonal IgMs derived from leukemic KL25 × Eμ-TCL1 and VI10YEN × Eμ-TCL1 mice were screened for autoreactivity by using an autoantigen proteomic microarray comprising 124 different antigens. Monoclonal IgMs

were tested at a concentration of 1 mg/ml. An isotype control (ThermoFisher, catalog number: 026800) was used as a negative control. Autoantigens microarrays were manufactured, hybridized and scanned by the Genomics & Microarray Core Facility at UT Southwestern Medical Center. Normalized fluorescent intensity values were analyzed with Cluster 3.0 and Java-TreeView software to generate the heat map. Color scale ranges between +200 and −200 standard deviations.

### Statistical analyses

Results are expressed as mean + SEM. All statistical analyses were performed in Prism (GraphPad Software). Means among three or more groups were compared with one-way or two-way analysis of variance with Bonferroni's post-test. Kaplan–Meier survival curves were compared with the log-rank (Mantel-Cox) test. Exact *P*-values for each experiment are reported in Appendix Table S1.

**Expanded View** for this article is available online.

### Acknowledgements

We thank M. Silva for secretarial assistance; M. Mainetti, L. Giustini, and M. Raso for technical assistance; C. Croce for Eμ-TCL1 mice; A. Mondino for critical reading of the manuscript; and all the members of the Iannacone laboratory for helpful discussions. Flow cytometry was carried out at FRACTAL, a flow cytometry resource and advanced cytometry technical applications laboratory established by the San Raffaele Scientific Institute. This work was supported by ERC grants 281648 and 725038 (to M.I.), Italian Association for Cancer Research (AIRC) grants 15350 (to M.I.), 15189 (to P.G.) and 9965 (to M.I., P.G. and F.C.C), Italian Ministry of Health grant GR-2011-02347925 (to M.I.), Fondazione Regionale per la Ricerca Biomedica grant 2015-0010 (to M.I.), European Molecular Biology Organization (EMBO) Young Investigator Program (to M.I.), and a Career Development Award from the Giovanni Armenise-Harvard Foundation (to M.I.).

### Author contributions

NJO and MDG designed and performed experiments, analyzed data, prepared the figures, and wrote the paper; JF, PDL, and AF performed experiments and analyzed data; RÜ performed the cell autonomous signaling experiments on TKO cells; SI performed the initial light-chain sequencing experiments; DGE, FC-C, HJ, PG, and LGG provided conceptual advice and revised the paper; MI designed and coordinated the study, provided funding, analyzed the data, and wrote the paper.

### Conflict of interest
The authors declare that they have no conflict of interest.

### For more information
https://www.lls.org/leukemia/chronic-lymphocytic-leukemia
http://cllsociety.org/
http://www.airc.it/
http://www.iannaconelab.com/

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
