## [Review Process File · EMBO Molecular Medicine]

Pathogen-specific B cell receptors drive chronic lymphocytic leukemia by light chain-dependent cross-reaction with autoantigens

Nereida Jiménez de Oya, Marco De Giovanni, Jessica Fioravanti, Rudolf Übelhart, Pietro Di Lucia, Amleto Fiocchi, Stefano Iacovelli, Dimitar G. Efremov, Federico Caligaris-Cappio, Hassan Jumaa, Paolo Ghia, Luca G. Guidotti and Matteo Iannacone

Corresponding author: Matteo Iannacone, IRCCS San Raffaele Scientific Institute

Review timeline:

Submission date:	22 February 2017
Editorial Decision:	05 April 2017
Revision received:	21 July 2017
Editorial Decision:	02 August 2017
Revision received:	11 August 2017
Accepted:	15 August 2017

Transaction Report:

Editor: Roberto Buccione

1st Editorial Decision

05 April 2017

Thank you for the submission of your manuscript to EMBO Molecular Medicine. We have now heard back from the Reviewers whom we asked to evaluate your manuscript.

I again apologise for the unusual delay in reaching a decision on your manuscript. In this case, we first experienced significant difficulties in securing expert and willing Reviewers. I eventually only managed to secure two reviewers. Further to this the evaluations were delivered with some delay.

I am therefore proceeding based on the two evaluations obtained so far as further delay cannot be justified and would not be productive.

As you will see, although the reviewers find your work potentially interesting and relevant and although Reviewer 2 is less reserved, both raise complementary and in part overlapping fundamental concerns. The basic issue is essentially the lack of sufficient support for the main claims, especially that autoreactive BCRs drive CLL progression.

Our reviewer cross-commenting exercise led to reviewer convergence on the issues raised and agreement that the main issue is that the link between autoantigen recognition by sera in leukemic mice and the antigen specificity of the BCRs selected during CLL development is missing. Reviewer 2 also agreed that indeed to prove this point, cloning of the BCRs selected during CLL development and defining the antigen specificities of the cloned BCRs, are required and that without

such experimental support, the manuscript cannot be published. I should add that reviewer 2 also agreed with reviewer 1 in raising the question as to whether CLL might not develop in the DH-LMP2A-E μ -Tcl-1 mice because the precursor cells for CLL are missing. Finally, it was also noted that both reviewers wondered why E μ -Tcl1 mice lacking transgenic Ig heavy chains were excluded from further analysis.

In conclusion, while publication of the paper cannot be considered at this stage, given the potential interest of your findings and after internal discussion, we have decided to give you the opportunity to address the criticisms.

We are thus prepared to consider a substantially revised submission, with the understanding that the Reviewers' concerns must be addressed with additional experimental data where appropriate and as outlined above, and that acceptance of the manuscript will entail a second round of review. The overall aim is to significantly upgrade the relevance and conclusiveness of the dataset, which of course is of paramount importance for our title.

Please note that it is EMBO Molecular Medicine policy to allow a single round of revision only and that, therefore, acceptance or rejection of the manuscript will depend on the completeness of your responses included in the next, final version of the manuscript.

Since as mentioned above, the required revision in this case appears to require a significant amount of time, additional work and experimentation, and might be technically challenging, I would understand if you chose to rather seek publication elsewhere at this stage. Should you do so, and we hope not, we would welcome a message to this effect.

EMBO Molecular Medicine now requires a complete author checklist (<http://embomolmed.embopress.org/authorguide#editorial3>) to be submitted with all revised manuscripts. Provision of the author checklist is mandatory at revision stage; the checklist is designed to enhance and standardize reporting of key information in research papers and to support reanalysis and repetition of experiments by the community. The list covers key information for figure panels and captions and focuses on statistics, the reporting of reagents, animal models and human subject-derived data, as well as guidance to optimise data accessibility. This checklist especially relevant in this case given the issues raised with respect to statistical treatment and animal numbers.

As you know, EMBO Molecular Medicine has a "scooping protection" policy, whereby similar findings that are published by others during review or revision are not a criterion for rejection. However, I do ask you to get in touch with us after three months if you have not completed your revision, to update us on the status. Please also contact us as soon as possible if similar work is published elsewhere.

Last, but not least, please carefully conform to our author guidelines (<http://embomolmed.embopress.org/authorguide>) to ensure rapid pre-acceptance processing in case of a favourable outcome on your revision.

I look forward to seeing a revised form of your manuscript in due time.

***** Reviewer's comments *****

Referee #1 (Comments on Novelty/Model System):

My main reservation re technical quality and novelty is that no direct evidence is presented supporting the claim that auto reactive BCRs drive CLL progression. This would require the analysis of BCRs cloned from CLL cells.

Referee #1 (Remarks):

This paper addresses an old, still not fully resolved question in CLL research, namely to which extent and through the recognition of which antigens the BCR expressed by CLL cells drives tumor

development and progression. The authors claim to resolve three issues: 1. BCR engagement is absolutely required for CLL development in their model (Abstract). 2. High-affinity (viral) antigen recognition does not affect CLL development or progression (p. 6, lines 13/14). 3. Pathogen-specific receptors drive CLL by light chain-dependent cross-reactions with autoantigens (Title). However, none of these claims is fully justified.

Ad 1: The authors use the Em-TCL1 mouse model in their experiments, in combination with various transgenic BCRs or a BCR mimic, the EBV LMP2A protein. In the case of the latter B cell development is driven by transgenic LMP2A (DHLMP2A), expressed in the cells instead of a BCR. Because CLL development is abolished in this situation, the authors make their argument on the absolute requirement of the BCR. However, in the Em-TCL1 model CLL develops from B1 cells, and the original paper on DHLMP2A mice showed that B1 cells were essentially missing in these animals. Surprisingly, the authors of the present paper seem to find (some) B1 cells in their DHLMP2A x Em-TCL1 cross; but they do not comment on this discrepancy and a possible involvement of the TCL1 transgene. As things stand, one cannot exclude that in mice expressing DHLMP2A instead of a BCR the true CLL progenitor cells are missing, and that this may cause the absence of CLL development rather than the absence of a BCR.

Ad 2: The argument about the failure of high-affinity antigen recognition to affect CLL development and progression is based on mouse crosses in which the Em-TCL1 transgene is combined with transgenic BCRs carrying specificity for either LCMV or VSV virus. The evidence presented is that neither infection of the mice with VSV weeks before CLL development nor three immunizations with an LCMV peptide before the onset of disease alter its course. (LCMV infection abolishes CLL development in both BCR transgenic and control animals, for unknown reasons.) Overall the evidence presented in this context is limited, negative and suggestive at most.

Ad 3: Here the IgL repertoire analysis presented by the authors and the finding of a striking range of autoantibody specificities in the mice carrying virus-specific BCRs and the Em-TCL1 transgene is interesting and suggestive along the lines of the authors' thinking; but the crucial experiments, namely the demonstration that the autoantibodies are indeed produced by the malignant cells, are missing. This has to be done the hard way, namely cloning BCRs from leukemic cells and testing them for auto reactivity.

Other, related points:

1) Fig. 3E: The CDR3 sequences should be taken out as they are truly misleading. And why are there no data on Em-TCL1 animals without transgenic BCRs? Would one not expect autoreactivity in that case as well? Is this perhaps already known and published?

2) Fig. S1B,C: Does the Em-TCL1 transgene cause CD5 up-regulation in B cells? Are B1 phenotype cells detectable in these animals already earlier in life? How do the authors explain the discrepancy between their data and the original Casola et al. paper on DHLMP2A mice, which were essentially devoid of B1 cells? Did they test these mice in their facility?

Referee #2 (Remarks):

In the manuscript "Pathogen-specific B cell receptors drive chronic lymphocytic leukemia by light chain-dependent cross-reaction with autoantigens" Nereida Jiménez de Oya et al. set out to study the role of B cell receptor (BCR) signaling in the development and progression of chronic lymphocytic leukemia (CLL) in the well established murine E μ -Tcl1 model system. To this end, the authors crossed transgenic E μ -Tcl1 mice with mice expressing in their B cells (i) Ig heavy chains with neutralizing specificity for LCMV (KL25), (ii) Ig heavy and light chains with neutralizing specificity for VSV (VI10YEN), and with mice (iii) carrying a targeted replacement of the IgH locus by the Epstein-Barr virus LMP2A gene. LMP2A provides a survival signal for B cells and the B cells lack surface-expressed and secreted immunoglobulins. In the absence of antigenic stimulation KL25 x E μ -Tcl1 mice developed CLL with the same kinetics as E μ -Tcl1 mice. VI10YEN x E μ -Tcl1 mice developed a much less aggressive leukemia, whereas CLL development was completely abolished in DH-LMP2A x E μ -Tcl1-mice indicating that the tonic signal provided by LMP2A is too weak to drive CLL development. Infection of VI10YEN x E μ -Tcl1 mice with VSV did not affect CLL development and progression, although antibody titers against VSV were

significantly increased upon infection with VSV. Surprisingly, infection of KL25 x E μ -Tc11 mice with LCMV prevented development of CLL. This particular point was excluded for a further detailed analysis in a separate manuscript. To avoid confounding effects brought about by virus infection, KL25 x E μ -Tc11 mice were instead immunized with LCMV glycoprotein (LCMV-GP). Also in this model high affinity antigen recognition - as revealed by rising antibody titers upon immunization - did not affect CLL development and progression.

In search for mechanisms unrelated to virus-specific BCR stimulation the authors analyzed the BCR repertoire of pre-leukemic and leukemic KL25 x E μ -Tc11- and V110YEN x E μ -Tc11-mice and compared it to age-matched KL25 and V110YEN controls lacking the E μ -Tc11 transgene. All analyzed leukemic mice expressed the transgenic Ig heavy chain as expected, whereas all leukemic mice had a biased light chain usage which might suggest selection of BCRs during CLL development that cross-react with one or several autoantigens. If such CLL-selected BCRs recognize an internal epitope of the BCR itself, this will lead to cell-autonomous signaling. To test for cell-autonomous signaling activity, the respective heavy and light chains were expressed in the BCR-deficient B cell line TKO which expresses an inactive BLNK adaptor-estrogen receptor fusion protein that is activated by the addition of 4-hydroxytamoxifen (4-OHT). Expression of the KL25 heavy chain together with two different light chains that were most frequently selected from leukemic CLL clones resulted in cell-autonomous signaling, whereas light chains expressed together with KL25 heavy chain from pre-leukemic mice did not. Yet, when expressed with the respective V110 heavy chain, light chains selected in leukemic V110YEN x E μ -Tc11 mice did not confer cell-autonomous signaling. The fact that light chains selected in leukemic V110YEN x E μ -Tc11 mice did not confer cell-autonomous signaling indicated that selection of antibodies with specificity for BCR epitopes is not the only pathogenetic mechanism involved in CLL development, and that leukemic BCRs might cross react with different autoantigens. Indeed, sera from leukemic KL25 x E μ -Tc11- and V110YEN x E μ -Tc11-mice reacted with a large panel of different autoantigens that are known to be targeted by autoantibodies in various autoimmune diseases as well as in CLL in men and mice. The data suggest that light chains with broad specificity for autoantigens that are selected after infection with viral pathogens drive CLL development by continuous encounter of a wide range of autoantigens.

This is a highly interesting story with high relevance for CLL development. The quality of the data presented is high, yet, there remain a number of questions that should be addressed experimentally and also dealt with in the discussion.

1. In Figure 3E sera from preleukemic and leukemic KL25 x E μ -Tc11- and V110YEN x E μ -Tc11-mice and KL25 and V110YEN control mice were used for recognition of a broad range of autoantigens. It is suggested but not proven that recognition of these autoantigens is brought about by the light chains selected during CLL development. It is not known from the data presented whether recognition of the broad range of autoantigens as shown in Figure 3E is mediated by many different antibodies (each of which may recognize a different autoantigen) or the few antibodies with the broad specificity for many autoantigens that are selected during CLL development. The link between antibody specificity in the sera and the antigen specificity of the BCRs on leukemic cells is still missing.

2. The two light chains selected most frequently during CLL development in KL25 x E μ -Tc11-mice have been shown to confer cell autonomous signaling upon addition of 4-OHT after transduction into TKO cells together with KL25 heavy chain (Figure 3C) indicating that the BCR recognizes an epitope on the BCR itself. On the other hand, sera from these leukemic mice were shown to recognize the same broad range of autoantigens as sera from leukemic V110YEN x E μ -Tc11-mice (Figure 3E). The finding that the selected light chains confer cell autonomous signaling in leukemic KL25 x E μ -Tc11-mice provides a pathogenetic mechanism for CLL development on its own and raises the question whether recognition of the broad range of autoantigens is an epiphenomenon or contributes additionally to CLL development in the way suggested for V110YEN x E μ -Tc11-mice.

3. Transgenic expression of a virus-specific Ig heavy chain had no impact on CLL development in KL25 x E μ -Tc11-mice and has reduced the aggressiveness of CLL to a significant extent in V110YEN x E μ -Tc11-mice. CLL development in E μ -Tc11-mice has been studied for comparison throughout this manuscript, but is unfortunately excluded in the experiments presented in Figure 3. Selection of specific heavy and light chains (figure 3A) and recognition of autoantigens by sera of

leukemic and preleukemic mice is equally interesting in this condition and may provide an answer to the question whether and how infections may shape the BCR repertoire for CLL development.

In summary, this is a highly interesting manuscript which should be considered for publication after careful revision.

1st Revision - authors' response

21 July 2017

Response to Reviewers comments on Jimenez De Oya N.*, De Giovanni M.* et al. "Pathogen-specific B cell receptors drive CLL via light chain dependent cross-reaction with autoantigens". (EMM-2017-00732)

We wish to thank the reviewers for the scholarly review of our work and the very helpful comments. Based on their constructive suggestions we have revised our manuscript and added substantial new experimental data that, in our opinion, positively address all the major and minor concerns raised, significantly improving the manuscript. Two main figures have been modified in response to the reviewers' comments. Two additional ones - termed Reviewer Figures 1 and 2 - have been included in this letter for the reviewers' benefit; while addressing specific comments, we believe that the data depicted in these latter figures remain tangential to the main messages of our work and, as such, they should not be incorporated in the final version.

Response to reviewer 1

"My main reservation are technical quality and novelty is that no direct evidence is presented supporting the claim that auto reactive BCRs drive CLL progression. This would require the analysis of BCRs cloned from CLL cells. This paper addresses an old, still not fully resolved question in CLL research, namely to which extent and through the recognition of which antigens the BCR expressed by CLL cells drives tumor development and progression. The authors claim to resolve three issues: 1. BCR engagement is absolutely required for CLL development in their model (Abstract). 2. High-affinity (viral) antigen recognition does not affect CLL development or progression (p. 6, lines 13/14). 3. Pathogen-specific receptors drive CLL by light chain-dependent cross-reactions with autoantigens (Title). However, none of these claims is fully justified.

Ad 1: The authors use the Em-TCL1 mouse model in their experiments, in combination with various transgenic BCRs or a BCR mimic, the EBV LMP2A protein. In the case of the latter B cell development is driven by transgenic LMP2A (DHLMP2A), expressed in the cells instead of a BCR. Because CLL development is abolished in this situation, the authors make their argument on the absolute requirement of the BCR. However, in the Em-TCL1 model CLL develops from B1 cells, and the original paper on DHLMP2A mice showed that B1 cells were essentially missing in these animals. Surprisingly, the authors of the present paper seem to find (some) B1 cells in their DHLMP2AxEm-TCL1 cross; but they do not comment on this discrepancy and a possible involvement of the TCL1 transgene. As things stand, one cannot exclude that in mice expressing DHLMP2A instead of a BCR the true CLL progenitor cells are missing, and that this may cause the absence of CLL development rather than the absence of a BCR.

We would like to point out that, in the original paper describing the D_HLMP2A mice (generated on a Balb/c background), Casola and coworkers detected a very significant reduction of CD5⁺ B cells in the peritoneum, but little or no difference in secondary lymphoid organs (see Figure 2A of Casola et al., Nat Immunol 2004). Our data – obtained in D_HLMP2A backcrossed for more than 10 generations with C57BL/6 mice, crossed to Em-TCL1 mice and bred in a difference mouse facility – are essentially in line with those obtained in the original publication (significant reduction of CD5⁺ B cells in the peritoneum, but no differences in secondary lymphoid organs). We believe that the absence of any detectable difference in the number of CD5⁺ B cells in the spleen of D_HLMP2A x Em-TCL1 mice is particularly relevant to this project, as the adoptive transfer of splenocytes from Em-TCL1 mice into WT recipients was shown to fully recapitulate the disease observed in Em-TCL1 mice (suggesting that the spleen contains the putative CLL precursor in this model {Bichi:2002fc}). Moreover, we observed reduction in the number of peritoneal CD5⁺ cells to the same extent to those observed in D_HLMP2A x Em-TCL1 mice in KL25 x Em-TCL1 mice (that develop CLL with the same kinetic of Em-TCL1 mice) and in VII10YEN x Em-TCL1 mice (that

develop CLL, albeit with a slower kinetic than Em-TCL1 mice). While these results are consistent with the notion that B cell fate is determined by signal strength, they also argue against the reduction in peritoneal CD5⁺ B cells observed in D_HLMP2A mice as a critical factor explaining the absence of CLL development in these mice.

Ad 2: The argument about the failure of high-affinity antigen recognition to affect CLL development and progression is based on mouse crosses in which the Em-TCL1 transgene is combined with transgenic BCRs carrying specificity for either LCMV or VSV virus. The evidence presented is that neither infection of the mice with VSV weeks before CLL development nor three immunizations with an LCMV peptide before the onset of disease alter its course. (LCMV infection abolishes CLL development in both BCR transgenic and control animals, for unknown reasons.) Overall the evidence presented in this context is limited, negative and suggestive at most”.

We agree with the reviewer that the evidence presented in this context is negative. However, we believe that the data provided are still rather informative: in experimental conditions where antigen-specific B cells were induced to proliferate and differentiate into Ab-secreting cells, infection or immunization did not alter the clinical course of CLL development. These results are consistent with previous correlative studies in mice {Iacovelli:2015kt} and suggest that high-affinity antigen recognition is not a major driver of CLL progression. Nonetheless, we have revised the text summarizing those results to avoid possible over interpretations of our results.

Ad 3: Here the IgL repertoire analysis presented by the authors and the finding of a striking range of autoantibody specificities in the mice carrying virus-specific BCRs and the Em-TCL1 transgene is interesting and suggestive along the lines of the authors' thinking; but the crucial experiments, namely the demonstration that the autoantibodies are indeed produced by the malignant cells, are missing. This has to be done the hard way, namely cloning BCRs from leukemic cells and testing them for auto reactivity.

We thank the reviewer for raising this critical point. To formally demonstrate that the autoantibodies were indeed produced by the malignant cells, we generated monoclonal IgMs bearing the KL25 heavy chain coupled to the IGKV12-44*01 F/IGKJ2*01 F light chain associated to the LCDR3 motif CQH-HYGTYPY-TF (the most frequently selected light chain in leukemic KL25 x E μ -TCL1 mice) or the V110 heavy chain coupled to the IGKV6-32*01 F/IGKJ2*01 F light chain associated to the LCDR3 motif CQQ-DYSS-TF (the most frequently selected light chain in leukemic V110YEN x E μ -TCL1 mice). Both monoclonal antibodies showed a significant degree of autoreactivity, with the antibody derived from KL25 x E μ -TCL1 mice recognizing a broader range of autoantigens (see new **Fig. 4B**). These results indicate that leukemic KL25 x E μ -TCL1 and V110YEN x E μ -TCL1 mice preferentially selected light chains that confer BCRs the capacity to cross-react with a broad range of autoantigens.

Other, related points:

1) Fig. 3E: The CDR3 sequences should be taken out as they are truly misleading. And why are there no data on Em-TCL1 animals without transgenic BCRs? Would one not expect autoreactivity in that case as well? Is this perhaps already known and published?

As suggested, we have removed the CDR3 sequences from the original **Fig. 3E** (new **Fig. 4A**). To assess whether polyclonal E μ -TCL1 mice produce autoantibodies, we tested sera from either 8 week-old pre-leukemic E μ -TCL1 mice or 9 month-old leukemic E μ -TCL1 mice for autoreactivity. Consistent with previously published data in mouse models and in patients {Yan:2006fp, Zhang:2014ir, Anonymous:2013ko, Stevenson:2014cn}, sera from leukemic E μ -TCL1 showed high level of reactivity against several different autoantigens (new **Fig. 4A**).

2) Fig. S1B,C: Does the Em-TCL1 transgene cause CD5 up-regulation in B cells? Are B1 phenotype cells detectable in these animals already earlier in life? How do the authors explain the discrepancy between their data and the original Casola et al. paper on DHLMP2A mice, which were essentially devoid of B1 cells? Did they test these mice in their facility?

Our data argue against TCL1 causing early CD5 upregulation in B cells, given that similar numbers of CD5⁺ B cell were found in WT and E μ -TCL1 mice at 8 weeks of age (**Fig. S1**). To test whether CD5⁺ B cells are detectable earlier in life in our cohorts of mice, we quantified the number of CD5⁺ B cells in 4 week-old animals. As shown in **Reviewer Fig. 1** below, CD5⁺ B cells are detectable in

all groups of mice, but the various groups of mice show differences comparable to the ones observed at 8 weeks of age (see Fig. S1).

As per the presumed discrepancies between our data and the ones obtained by Casola et al., we would like again to point out that, in the original paper describing the D_HLMP2A mice (generated on a Balb/c background), Casola and coworkers detected a very significant reduction of CD5⁺ B cells in the peritoneum, but little or no difference in secondary lymphoid organs (see Figure 2A of {Casola:2004ed}). Our data – obtained in D_HLMP2A backcrossed for more than 10 generations with C57BL/6 mice, crossed to Em-TCL1 mice and bred in a difference mouse facility – are essentially in line with those obtained in the original publication (significant reduction of CD5⁺ B cells in the peritoneum, but no differences in secondary lymphoid organs, see Fig. S1).

Reviewer Figure 1

Reviewer Figure 1. Absolute numbers of CD19⁺ CD5⁺ cells in the blood, peritoneum, liver, inguinal lymph nodes (ing LNs) and spleen of 4-week old WT, E μ -TCL1, KL25 x E μ -TCL1, VII10YEN x E μ -TCL1 and D_HLMP2A x E μ -TCL1 mice.

Response to Reviewer 2

Referee #2 (Remarks):

In the manuscript "Pathogen-specific B cell receptors drive chronic lymphocytic leukemia by light chain-dependent cross-reaction with autoantigens" Nereida Jiménez de Oya et al. set out to study the role of B cell receptor (BCR) signaling in the development and progression of chronic lymphocytic leukemia (CLL) in the well established murine E μ -Tcl1 model system. To this end, the authors crossed transgenic E μ -Tcl1 mice with mice expressing in their B cells (i) Ig heavy chains with neutralizing specificity for LCMV (KL25), (ii) Ig heavy and light chains with neutralizing specificity for VSV (VII10YEN), and with mice (iii) carrying a targeted replacement of the IgH locus by the Epstein-Barr virus LMP2A gene. LMP2A provides a survival signal for B cells and the B cells lack surface-expressed and secreted immunoglobulins. In the absence of antigenic stimulation KL25 x E μ -Tcl1 mice developed CLL with the same kinetics as E μ -Tcl1 mice. VII10YEN x E μ -Tcl1 mice developed a much less aggressive leukemia, whereas CLL development was completely abolished in D_H-LMP2A x E μ -Tcl1-mice indicating that the tonic signal provided by LMP2A is too weak to drive CLL development. Infection of VII10YEN x E μ -Tcl1 mice with VSV did not affect CLL development and progression, although antibody titers against VSV were significantly increased upon infection with VSV. Surprisingly, infection of KL25 x E μ -Tcl1 mice with LCMV prevented development of CLL. This particular point was excluded for a further detailed analysis in a separate manuscript. To avoid confounding effects brought about by virus infection, KL25 x E μ -Tcl1 mice were instead immunized with LCMV glycoprotein (LCMV-GP). Also in this model high affinity antigen recognition - as revealed by rising antibody titers upon immunization - did not affect CLL development and progression. In search for mechanisms unrelated to virus-specific BCR stimulation the authors analyzed the BCR repertoire of pre-leukemic and leukemic KL25 x E μ -Tcl1- and VII10YEN x E μ -Tcl1-mice and compared it to age-matched KL25 and VII10YEN controls lacking the

Eμ-Tcl1 transgene. All analyzed leukemic mice expressed the transgenic Ig heavy chain as expected, whereas all leukemic mice had a biased light chain usage which might suggest selection of BCRs during CLL development that cross-react with one or several autoantigens. If such CLL-selected BCRs recognize an internal epitope of the BCR itself, this will lead to cell-autonomous signaling. To test for cell-autonomous signaling activity, the respective heavy and light chains were expressed in the BCR-deficient B cell line TKO which expresses an inactive BLNK adaptor-estrogen receptor fusion protein that is activated by the addition of 4-hydroxytamoxifen (4-OHT). Expression of the KL25 heavy chain together with two different light chains that were most frequently selected from leukemic CLL clones resulted in cell-autonomous signaling, whereas light chains expressed together with KL25 heavy chain from pre-leukemic mice did not. Yet, when expressed with the respective VII0 heavy chain, light chains selected in leukemic VII0YEN x Eμ-Tcl1 mice did not confer cell-autonomous signaling. The fact that light chains selected in leukemic VII0YEN x Eμ-Tcl1 mice did not confer cell-autonomous signaling indicated that selection of antibodies with specificity for BCR epitopes is not the only pathogenetic mechanism involved in CLL development, and that leukemic BCRs might cross react with different autoantigens. Indeed, sera from leukemic KL25 x Eμ-Tcl1- and VII0YEN x Eμ-Tcl1-mice reacted with a large panel of different autoantigens that are known to be targeted by autoantibodies in various autoimmune diseases as well as in CLL in men and mice. The data suggest that light chains with broad specificity for autoantigens that are selected after infection with viral pathogens drive CLL development by continuous encounter of a wide range of autoantigens. This is a highly interesting story with high relevance for CLL development. The quality of the data presented is high, yet, there remain a number of questions that should be addressed experimentally and also dealt with in the discussion.

1. In Figure 3E sera from preleukemic and leukemic KL25 x Eμ-Tcl1- and VII0YEN x Eμ-Tcl1-mice and KL25 and VII0YEN control mice were used for recognition of a broad range of autoantigens. It is suggested but not proven that recognition of these autoantigens is brought about by the light chains selected during CLL development. It is not known from the data presented whether recognition of the broad range of autoantigens as shown in Figure 3E is mediated by many different antibodies (each of which may recognize a different autoantigen) or the few antibodies with the broad specificity for many autoantigens that are selected during CLL development. The link between antibody specificity in the sera and the antigen specificity of the BCRs on leukemic cells is still missing.

We thank the reviewer for the overall positive assessment of our work and for raising this very valid point. To formally demonstrate that the autoantibodies were indeed produced by the malignant cells, we generated monoclonal IgMs bearing the KL25 heavy chain coupled to the IGKV12-44*01 F/IGKJ2*01 F light chain associated to the LCDR3 motif CQH-HYGTPY-TF (the most frequently selected light chain in leukemic KL25 x Eμ-TCL1 mice) or the VII0 heavy chain coupled to the IGKV6-32*01 F/IGKJ2*01 F light chain associated to the LCDR3 motif CQQ-DYSS-TF (the most frequently selected light chain in leukemic VII0YEN x Eμ-TCL1 mice). Both monoclonal antibodies showed a significant degree of autoreactivity, with the antibody derived from KL25 x Eμ-TCL1 mice recognizing a broader range of autoantigens (see new **Fig. 4B**). These results indicate that leukemic KL25 x Eμ-TCL1 and VII0YEN x Eμ-TCL1 mice preferentially selected light chains that confer BCRs the capacity to cross-react with a broad range of autoantigens.

2. The two light chains selected most frequently during CLL development in KL25 x Eμ-Tcl1-mice have been shown to confer cell autonomous signaling upon addition of 4-OHT after transduction into TKO cells together with KL25 heavy chain (Figure 3C) indicating that the BCR recognizes an epitope on the BCR itself. On the other hand, sera from these leukemic mice were shown to recognize the same broad range of autoantigens as sera from leukemic VII0YEN x Eμ-Tcl1-mice (Figure 3E). The finding that the selected light chains confer cell autonomous signaling in leukemic KL25 x Eμ-Tcl1-mice provides a pathogenetic mechanism for CLL development on its own and raises the question whether recognition of the broad range of autoantigens is an epiphenomenon or contributes additionally to CLL development in the way suggested for VII0YEN x Eμ-Tcl1-mice.

We would like to point out that autonomous signaling activity indicates that the BCR is recognizing one or more autoantigens displayed by TKO cells (the internal epitope of the BCR being only one of potentially many said autoantigens). Our findings indicate that leukemic mice preferentially select light chains that confer autoreactivity (including, in case of KL25 x Eμ-TCL1 mice, autonomous signaling activity; incidentally, this is the first report of specific light chains conferring a heavy chain the ability to become autonomously active).

The relative role of the different autoantigens (internal epitope of the BCR versus the many autoantigens described here) in the pathogenesis of CLL remains to be determined and it will require the generation of additional mouse models. In this regard, it will be interesting to investigate whether the antigens that were recognized consistently by all leukemic sera (and by the monoclonal IgMs derived from leukemic mice) play a particularly important role in CLL pathogenesis.

3. *Transgenic expression of a virus-specific Ig heavy chain had no impact on CLL development in KL25 x E μ -Tcl1-mice and has reduced the aggressiveness of CLL to a significant extent in VII0YEN x E μ -Tcl1-mice. CLL development in E μ -Tcl1-mice has been studied for comparison throughout this manuscript, but is unfortunately excluded in the experiments presented in Figure 3. Selection of specific heavy and light chains (figure 3A) and recognition of autoantigens by sera of leukemic and preleukemic mice is equally interesting in this condition and may provide an answer to the question whether and how infections may shape the BCR repertoire for CLL development. In summary, this is a highly interesting manuscript, which should be considered for publication after careful revision.*

As suggested, we have now included E μ -TCL1 mice in the experiments reported in the original **Fig. 3**. Specifically, we have now assessed whether polyclonal E μ -TCL1 mice produce autoantibodies by testing sera from either 8 week-old pre-leukemic E μ -TCL1 mice or 9 month-old leukemic E μ -TCL1 mice for autoreactivity. Consistent with previously published data in mouse models and in patients {Yan:2006fp, Zhang:2014ir, Anonymous:2013ko, Stevenson:2014cn}, sera from leukemic E μ -TCL1 mice showed high level of reactivity against several different autoantigens (new **Fig. 4A**). Furthermore, we cloned and sequenced the light chains from four 9-month old leukemic E μ -TCL1 mice. As shown in **Reviewer Fig. 2**, the most represented light chains in leukemic polyclonal E μ -TCL1 showed a high degree of similarity to light chains described in anti-DNA/ANA antibodies, with two light chains being previously described in the E μ -TCL1 model {Yan:2006fp}. Together, the data are consistent with the hypothesis that polyclonal E μ -TCL1 mice preferentially select light chains that confer BCRs the capacity to cross-react with a broad range of autoantigens.

Reviewer Figure 2

Reviewer Figure 2. Pie chart representing the light chain repertoire in 9 month-old leukemic E μ -TCL1 mice. $n=4$.

Thank you for the submission of your revised manuscript to EMBO Molecular Medicine. We have now received the enclosed reports from the reviewers that were asked to re-assess it. As you will see the reviewers are now supportive, although both have a few final and quite important requests that require your action.

I am prepared to accept your manuscript for publication pending satisfactory compliance with the reviewers' final requests. I am also prepared to make an editorial decision on your revised manuscript provided you carefully deal with the final criticisms. Please also fulfil the following editorial requirements:

- 1) Please provide 5 keywords, a running title and a conflict of interest statement.
- 2) The supplementary tables and figures would be best made into expanded view (EV) tables and figures (<http://embomolmed.embopress.org/authorguide#expandedview>). As a consequence, nomenclature and appropriate callouts in the manuscript should be carefully amended.
- 3) During our pre-acceptance figure-checking routines, we noticed that Fig 1A and S3 present some identical data points (red), albeit but paired with other data. Please explain this occurrence, and make sure this is clarified in the figure legends.
- 4) You have chosen the Report format for your manuscript for which however, only three main figures are allowed. Please choose one to become an EV figure (possibly number 1) and as mentioned above, nomenclature and appropriate callouts in the manuscript should be carefully amended.
- 5) We are still missing precise information on the gender and age of the mice used in the various experimental settings. Please update both the manuscript and the checklist to reflect this information.
- 6) As per our Author Guidelines, the description of all reported data that includes statistical testing must state the name of the statistical test used to generate error bars and P values, the number (n) of independent experiments underlying each data point (not replicate measures of one sample), and the actual P value for each test (not merely 'significant' or 'P < 0.05'). Should you feel that inclusion of the P values (in legends or figures) impairs readability, you may opt to prepare an additional table displaying them, to be appropriately referred to in the figure legends and text.
- 7) We encourage the publication of source data, with the aim of making primary data more accessible and transparent to the reader. Would you be willing to provide a PDF file per figure that contains the original, uncropped and unprocessed scans of all or at least the key gels used in the manuscript and/or source data sets for relevant graphs? The files should be labelled with the appropriate figure/panel number, and in the case of gels, should have molecular weight markers; further annotation may be useful but is not essential. The files will be published online with the article as supplementary "Source Data" files. If you have any questions regarding this just contact me.
- 8) Every published paper includes a 'Synopsis' to further enhance discoverability. Synopses are displayed on the journal webpage and are freely accessible to all readers. They include a short description as well as 2-5 one-sentence bullet points that summarise the key NEW findings of the paper. The bullet points should be designed to be complementary to the abstract - i.e. not repeat the same text. We encourage inclusion of key acronyms and quantitative information. Please use the passive voice. Please attach this information in a separate file or send them by email, we will incorporate it accordingly. We also encourage the provision of striking image or visual abstract to illustrate your article. If you do, please provide a jpeg file 550 px-wide x 400-px high.

Please submit your revised manuscript within two weeks. I look forward to seeing a revised form of your manuscript as soon as possible.

***** Reviewer's comments *****

Referee #1 (Remarks):

The authors have substantially improved their paper by showing that the CLL cells really express autoreactive BCRs. I also will refrain from arguing further about the presence or absence of CLL progenitors in the DHLMP2A mice, although I still have my reservations about the strength of the argument of the authors, namely that all that matters is the absence or presence of a BCR. ("Absolutely required" (end of Discussion) - I would be more cautious.)

Here is something I want to leave at the authors' discretion: On pp 6/7 they write: "Together, these results suggest that high-affinity antigen recognition does not affect CLL development or progression, and they prompted us to investigate whether virus-specific BCRs may drive CLL pathogenesis by mechanisms that are unrelated to pathogen specificity." Do they want to say that the autoantigens in question are recognized through LOW affinity? Is there something else they are thinking about (pathogens versus autoantigens)? I would hope for some clarification in the final paper.

Referee #2 (Remarks):

Nereida Jiménez de Oya et al. have presented the revised version of the manuscript "Pathogen-specific B cell receptors drive chronic lymphocytic leukemia by light chain-dependent cross-reaction with autoantigens". They have carefully responded to all arguments of the reviewers. Importantly, they have performed the critical experiment that had been requested by both reviewers, namely to determine the spectrum of antigens recognized by the cloned B cell receptors of KL25 x E μ -Tc11 and VII0YEN x E μ -Tc11 mice and to compare these with the spectrum of antigens recognized by the sera of the respective mice. The data are presented in the novel Figure 4.

The data support the conclusion that the light chains pairing to a pathogen-specific heavy chain are selected during CLL development to form a BCR that recognizes a broad spectrum of autoantigens. Given that several clones evolve and predominate during CLL development it is not surprising that more autoantigens are recognized by the sera of leukemic mice than by a single (predominant) BCR of the respective leukemic mouse line.

The data are presented in Figure 4 as heat maps. This is o.k., but unfortunately, important information is not easily apparent and remains buried in the Excel file of supplementary Table 3. The reader would like to see how many and which autoantigens are recognized by the sera as well as the predominant BCR of a given leukemic mouse line, and even more importantly, which autoantigens recognized by the BCR of leukemic mice are not recognized by the sera of the respective mice.

Most of the autoantigens recognized by the leukemic sera are shared between leukemic KL25 x E μ -Tc11 and VII0YEN x E μ -Tc11 mice. Likewise, two thirds of the autoantigens recognized by the predominant BCRs of leukemic KL25 x E μ -Tc11 and VII0YEN x E μ -Tc11 mice are also shared. Given the more aggressive nature of CLL in KL25xEmu-Tc11 mice, it is not surprising that the BCR of leukemic KL25 x E μ -Tc11 mice recognizes about twice as many autoantigens as the BCR of leukemic VII0YEN x E μ -Tc11 mice. It is more difficult to reconcile, though, that a considerable part of the autoantigens recognized by the predominant BCR of leukemic KL25 x E μ -Tc11 mice (about one fourth) or by the predominant BCR of leukemic VII0YEN x E μ -Tc11 mice (about one third) is not recognized by the sera of the respective mice. The quantitative aspect of the data and this discrepancy should be dealt with in the Discussion. This information could be either included into an additional supplementary Figure presenting the same data in a different fashion, i.e. the order of autoantigens in Figure 4B should be identical to the one in Figure 4A, or alternatively, this information could be compiled in a supplementary table.

Editorial Requirements:

1) Please provide 5 keywords, a running title and a conflict of interest statement.

We have now provided keywords, running title and conflict of interest statement.

2) The supplementary tables and figures would be best made into expanded view (EV) tables and figures (<http://embomolmed.embopress.org/authorguide#expandedview>). As a consequence, nomenclature and appropriate callouts in the manuscript should be carefully amended.

We have now changed nomenclature and appropriate callouts in the manuscript.

3) During our pre-acceptance figure-checking routines, we noticed that Fig 1A and S3 present some identical data points (red), albeit but paired with other data. Please explain this occurrence, and make sure this is clarified in the figure legends.

Indeed, the same uninfected WT, Em-TCL1 and KL25 x Em-TCL1 controls were used in **Figure 1A** and **Figure EV3**. This is now explained in the legend to **Figure EV3**.

4) You have chosen the Report format for your manuscript for which however, only three main figures are allowed. Please choose one to become an EV figure (possibly number 1) and as mentioned above, nomenclature and appropriate callouts in the manuscript should be carefully amended.

As per Referee #2's suggestion, we have now merged original **Figure 4A** and **4B** into a single heat map that fits in **Figure 3** (see new **Figure 3E**). As such, we now fulfill the requirement for three main figures.

5) We are still missing precise information on the gender and age of the mice used in the various experimental settings. Please update both the manuscript and the checklist to reflect this information.

The gender and age of the mice used in the various experimental settings is now provided in the Figure legends and in the checklist.

6) As per our Author Guidelines, the description of all reported data that includes statistical testing must state the name of the statistical test used to generate error bars and P values, the number (n) of independent experiments underlying each data point (not replicate measures of one sample), and the actual P value for each test (not merely 'significant' or ' $P < 0.05$ '). Should you feel that inclusion of the P values (in legends or figures) impairs readability, you may opt to prepare an additional table displaying them, to be appropriately referred to in the figure legends and text.

We have prepared an additional table displaying the actual P value as well as the statistical tests used to generate error bars and P values.

7) We encourage the publication of source data, with the aim of making primary data more accessible and transparent to the reader. Would you be willing to provide a PDF file per figure that contains the original, uncropped and unprocessed scans of all or at least the key gels used in the manuscript and/or source data sets for relevant graphs? The files should be labelled with the appropriate figure/panel number, and in the case of gels, should have molecular weight markers; further annotation may be useful but is not essential. The files will be published online with the article as supplementary "Source Data" files. If you have any questions regarding this just contact me.

We have now provided a PDF file per figure that contains the source data sets (with the exception of **Figure 3** where the source data are already available as **Tables EV1-3**).

8) Every published paper includes a 'Synopsis' to further enhance discoverability. Synopses are

displayed on the journal webpage and are freely accessible to all readers. They include a short description as well as 2-5 one-sentence bullet points that summarise the key NEW findings of the paper. The bullet points should be designed to be complementary to the abstract - i.e. not repeat the same text. We encourage inclusion of key acronyms and quantitative information. Please use the passive voice. Please attach this information in a separate file or send them by email, we will incorporate it accordingly. We also encourage the provision of striking image or visual abstract to illustrate your article. If you do, please provide a jpeg file 550 px-wide x 400-px high.

We have now provided both a 'Synopsis' as well as a visual abstract.

Referee #1 (Remarks):

The authors have substantially improved their paper by showing that the CLL cells really express autoreactive BCRs. I also will refrain from arguing further about the presence or absence of CLL progenitors in the DHLMP2A mice, although I still have my reservations about the strength of the argument of the authors, namely that all that matters is the absence or presence of a BCR. ("Absolutely required" (end of Discussion) - I would be more cautious.)

We thank the reviewer for the positive assessment of our revised manuscript. As suggested, we have softened our final statement regarding the requirement of BCR expression for leukemia development.

Here is something I want to leave at the authors' discretion: On pp 6/7 they write: "Together, these results suggest that high-affinity antigen recognition does not affect CLL development or progression, and they prompted us to investigate whether virus-specific BCRs may drive CLL pathogenesis by mechanisms that are unrelated to pathogen specificity." Do they want to say that the autoantigens in question are recognized through LOW affinity? Is there something else they are thinking about (pathogens versus autoantigens)? I would hope for some clarification in the final paper.

We thank the reviewer for pointing this out. We have now better clarified this in our revised manuscript.

Referee #2 (Remarks):

Nereida Jiménez de Oya et al. have presented the revised version of the manuscript "Pathogen-specific B cell receptors drive chronic lymphocytic leukemia by light chain-dependent cross-reaction with autoantigens". They have carefully responded to all arguments of the reviewers. Importantly, they have performed the critical experiment that had been requested by both reviewers, namely to determine the spectrum of antigens recognized by the cloned B cell receptors of KL25 x E μ -Tcl1 and VII0YEN x E μ -Tcl1 mice and to compare these with the spectrum of antigens recognized by the sera of the respective mice. The data are presented in the novel Figure 4.

The data support the conclusion that the light chains pairing to a pathogen-specific heavy chain are selected during CLL development to form a BCR that recognizes a broad spectrum of autoantigens. Given that several clones evolve and predominate during CLL development it is not surprising that more autoantigens are recognized by the sera of leukemic mice than by a single (predominant) BCR of the respective leukemic mouse line.

The data are presented in Figure 4 as heat maps. This is o.k., but unfortunately, important information is not easily apparent and remains buried in the Excel file of supplementary Table 3. The reader would like to see how many and which autoantigens are recognized by the sera as well as the predominant BCR of a given leukemic mouse line, and even more importantly, which autoantigens recognized by the BCR of leukemic mice are not recognized by the sera of the respective mice.

Most of the autoantigens recognized by the leukemic sera are shared between leukemic KL25 x E μ -Tcl1 and VII0YEN x E μ -Tcl1 mice. Likewise, two thirds of the autoantigens recognized by the predominant BCRs of leukemic KL25 x E μ -Tcl1 and VII0YEN x E μ -Tcl1 mice are also shared.

Given the more aggressive nature of CLL in KL25xEmu-Tcl1 mice, it is not surprising that the BCR of leukemic KL25 x E μ -Tcl1 mice recognizes about twice as many autoantigens as the BCR of leukemic VII0YEN x E μ -Tcl1 mice. It is more difficult to reconcile, though, that a considerable part of the autoantigens recognized by the predominant BCR of leukemic KL25 x E μ -Tcl1 mice (about one fourth) or by the predominant BCR of leukemic VII0YEN x E μ -Tcl1 mice (about one third) is not recognized by the sera of the respective mice. The quantitative aspect of the data and this discrepancy should be dealt with in the Discussion. This information could be either included into an additional supplementary Figure presenting the same data in a different fashion, i.e. the order of autoantigens in Figure 4B should be identical to the one in Figure 4A, or alternatively, this information could be compiled in a supplementary table.

We thank the reviewer for the positive assessment of our revised manuscript and for raising this valid point! As suggested, we have combined original **Figure 4A** and **4B** in a single heat map (new **Figure 3E**). We hope that this will make it easier to compare the autoantigens recognized by the leukemic BCRs versus the ones recognized by the leukemic mouse sera. Of note, several reasons might explain the non-complete overlap between the autoantigens recognized by the monoclonal IgMs and the ones recognized by the leukemic sera and they include the presence of multiple malignant clones in leukemic mice as well as a different concentration of immunoglobulins in leukemic mouse sera compared to our monoclonal IgM preparations.

Corresponding Author Name: Matteo Iannacone

Journal Submitted to: Embo Molecular Medicine

Manuscript Number: 2017-0732